# Research on dynamic characteristics of large deformation shearer cable based on absolute node coordinate formulation method

Lijuan Zhao[1,2,3], Haining Zhang[1]*, Feng Gao[4], Liguo Han[1], Man Ge[5]

**1** School of Mechanical Engineering, Liaoning Technical University, Fuxin, China, **2** The State Key Lab of Mining Machinery Engineering of Coal Industry, Liaoning Technical University, Fuxin, China, **3** Liaoning Province Large Scale Industrial and Mining Equipment Key Laboratory, Fuxin, China, **4** Shandong Yankuang Group Changlong Cable Manufacturing Co., Ltd, Jining, China, **5** Shandong Yankuang Intelligent Manufacturing Co., Ltd, Jinan, China

* jscqtzhn@126.com

**Data Availability Statement:** Tensile test data of strands are available from the Science Data bank database. (URL: https://doi.org/10.57760/sciencedb.06896).

## Abstract

The development of intelligent and unmanned coal mining has put forward higher requirements on the service life and dynamic reliability of shearer cables. However, it is difficult to comprehensively consider the complexity of hosting conditions of coal mining working face and the dynamic characteristics of cables in different towing systems in the design and development of cables. The cables are periodized by pitch and have the same cross-sectional structure and properties. Based on the homogenization theory and volume average principle, the cable was assumed to be an orthotropic elastomer, and the tensile experimental method and finite element method were combined to calibrate the cable equivalent mechanical parameters. Based on the Absolute Node Coordinate Formulation (ANCF) method, the rigid-flexible coupled virtual prototype co-simulation model of shearer cable towing system was constructed to obtain the kinetic and kinematic parameters of each node of the cable and study the dynamic gradual change characteristics of the cable in different working areas. This research method has an important theoretical significance and engineering application value for the acquisition of dynamic characteristic parameters of shearer cables and the optimal design and dynamic reliability of cables.

## 1 Introduction

The research object in this paper is the mobile rubber-sheathed cables used in coal mine, which produce large deformation in following the shearer for many short-distance round trips. The cable bears complex and severe loads when working, and will be frequently subjected to various stresses such as bending, dragging, squeezing, twisting and possible impact of coal blocks or falling rocks, which may lead to mechanical damage, and may even cause accidents such as short circuit and grounding of the cable, reducing the service life of the cable and affecting the normal operation and safety production of the shearer. The traditional finite element model based on modal superposition cannot analyze the large deformation motion of

**Funding:** This work was supported by the National Natural Science Foundation of China [Grant number 51674134]. The funders had no role in study design, data collection and analysis, decision to publish, or preparation of the manuscript.

shearer cables in the towing system [1]. Shabana [2] firstly proposed the Absolute Nodal Coordinates Formulation (ANCF). The ANCF method takes the position and slope of each node in the global coordinate system as generalized coordinates, the mass matrix is a constant array, and the generalized force expression is simple, which has a greater advantage in solving the nonlinear large deformation problem of shearer cables. Fan et al. [3] introduced the elastic line method and verified that the elastic line method can effectively improve the computational efficiency with guaranteed accuracy by simulating the dynamics of a flexible pendulum. Based on the idea of elastic thin rod, You et al. [4] established a large deformation flexible cable dynamics model to analyze its mechanical properties considering spatial constraints, self-weight, dragging and friction effects, which provided a theoretical basis for cable path planning. Based on the Hertz contact collision theory, Yu et al. [5] proposed the expressions for the calculation of collision force and realized the collision simulation between flexible body and flexible body and flexible body and rigid body based on ANCF. Nikula et al. [6] discussed the usability of several beam elements based on the ANCF under torsion and double moment load scenarios. The higher-order element has some limitations under the double moment load conditions, and which should further be discretized. Du et al. [7] proposed a computer-aided flexible cable optimization design method based on dynamic analogy modeling. Combined with Cosserat theory and the principle of minimum potential energy, a nonlinear optimization model was established to simulate the digital wiring module. Wang et al. [8] discussed different numerical integration algorithms to solve the rigid and flexible cable system, and proposed an integration strategy combining implicit Euler and minimum residual method. The above scholars expanded the application of ANCF by theoretical calculation and explored the solving of the dynamics problem of large deformation flexible multibody system. When the ANCF method is used to study the large deformation motion of shearer cables, the complexity of hosting conditions of coal mining working face and the real boundary conditions of the cables in different towing systems should also be considered.

The cable is a complex spiral structure formed by multiple materials. It is difficult to analyze its dynamics simulation directly, so the mechanical properties of the cable need to be equivalent. In view of the equivalence of cables, Wang et al. [9] compared the stress-strain laws of different structural units inside the optical fiber composite submarine cable through the tensile test and finite element simulation, providing a reference for judging the working state and mechanical performance protection of the cable. Li [10] conducted a fundamental mechanical property research on single-stage and two-stage spiral strand structures, and concluded that the equivalent elastic modulus of the strand structure decreased with decreasing helix angle, and the friction and contact deformation had a large influence on the overall mechanical behavior. Michael et al. [11] proposed the volume average principle, which was used to predict the equivalent physical parameters of the model in a certain direction, using the average value of the stresses and strains in each element to equivalent the stress-strain level of the whole model. On the basis of traditional composite material theory, Wang et al. [12, 13] made various hypothetical theoretical predictions of the equivalent elastic modulus of superconducting busbar for ITER, and analyzed the equivalent elastic modulus of the cable by using homogenization theory. Wang et al. [14] determined the optimal number of layers of the cable insulation layer, inner sheath and outer sheath, which provided a theoretical basis for the temperature prediction and fault early warning of the mine power cable core. On the basis of effective Young's modulus and radius of curvature, Guo et al. [15] obtained the bending stiffness of multistage superconducting ITER cable, and which could be optimized by adjusting the spiral spacing of the cable. Combining tensile test and finite element method, the above scholars investigated the axial tensile properties of cables without large displacement and deformation, and provided a reference for the prediction of the equivalent elastic modulus of cables.

However, in order to obtain the equivalent mechanical parameters of the shearer cable with large deformation, the shear and torsion of the cable should also be considered.

Intelligent and unmanned shearer cables towing system is also an important part of intelligent coal mining [16, 17]. Our project team established high-precision 3D models of complex coal seam based on the DEM-FMBD, and collected vibration signals of shearer cutting process, and applied DCGAN-RFCNN network model to sense and identify coal cutting state dynamically. Based on the dynamic recognition of coal cutting state, the adaptive coal cutting system model integrated with machine-electric-hydraulic-control was constructed, and studied its adaptive cutting control strategy, which provided a research basis for analyzing the dynamic characteristics of cables in different towing systems in this paper [18–20]. The development of intelligent cable towing systems began with the Kopex intelligent working face in Poland, which integrated an automatic cable towing system to enable largely unmanned cable towing. Hao [21] designed the mechanical devices of the towing cable system and performed finite element analysis on chains to derive the Mises stress distribution laws under three kinds of loads. Cui et al. [22, 23] designed shearer cable automatic towing system based on circular chains drive and PLC variable frequency control, which realized the adaptive control of cable towing system. Li et al. [24] designed a shearer cable pulling force overload protection device to deal with cable damages due to shearer tension overload. The above scholars focused on the design of mechanical devices, strength analysis and synchronous control strategy, which had certain limitations. In addition, the study subject cable should also be put into the towing system, and its motion characteristics in different towing systems should be evaluated comprehensively according to the kinetics and kinematic parameters.

In this paper, 3D accurate modeling of shearer cable was constructed by considering manufacturing process and mathematical principle of spiral structure. Equivalent parameters of the smallest unit of copper strands for modeling analysis were calibrated by the tensile test. The cross-sectional structure and properties of the cable are the same and periodic. Based on the homogenization theory and volume average principle, the cable was assumed to be an orthotropic elastomer, and the equivalent elastic parameters of the three elastic symmetry surfaces were calibrated and verified by the finite element method. Based on the ANCF method, the rigid-flexible coupled virtual prototype co-simulation model of shearer cable towing system was constructed to obtain the kinetic and kinematic parameters of each node of the cable and study the dynamic gradual change characteristics of the cable in different working areas. This research method has an important theoretical significance and engineering application value for the acquisition of dynamic characteristic parameters of shearer cables and the optimal design and dynamic reliability of cables. The mind map of this paper is shown in Fig 1.

## 2 Construction of 3D geometrically accurate model of the cable

Take MCPT-1.9/3.3 KV shearer cable as the engineering object, and its cross-sectional structure is shown in Fig 2. Manufacturing processes are shown from Fig 3A to 3D, 3D accurate models are shown from Fig 3E to 3F. Several thinned copper wires are twisted into strands, which are the minimum units of 3D modeling analysis, as shown in Fig 3A and 3E. Multiple strands are twisted into conductors by cage stranding machine, as shown in Fig 3B and 3F. The ground conductor is placed in the center of the cable, and the control conductor is formed as the fourth conductor together with three power conductors to form a double spiral structure, as shown in Fig 3C and 3G. After wrapping, the neoprene sheath is extruded outside the cable to form the final cable product, as shown in Fig 3D and 3H. The accuracy and rationality of the 3D geometric accurate modeling are ensured by comprehensively considering the manufacturing process and the mathematical parameter equations of the spiral structures.

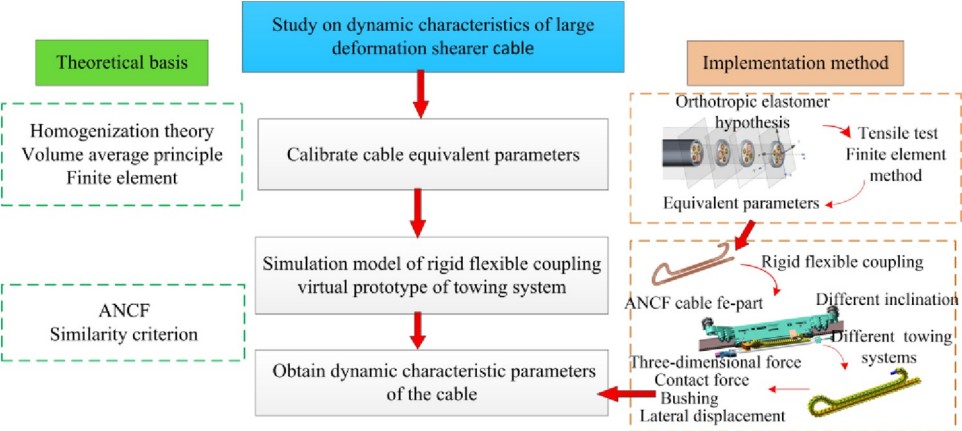

**Fig 1. Mind map.**

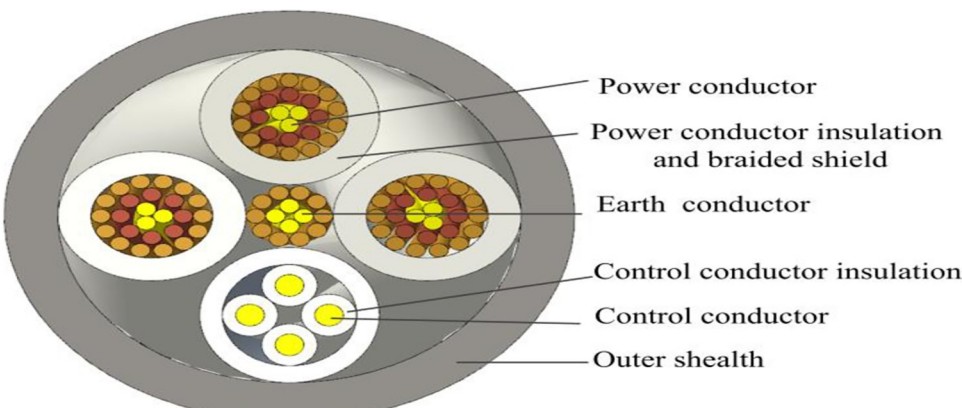

**Fig 2. Schematic section of shearer cable.**

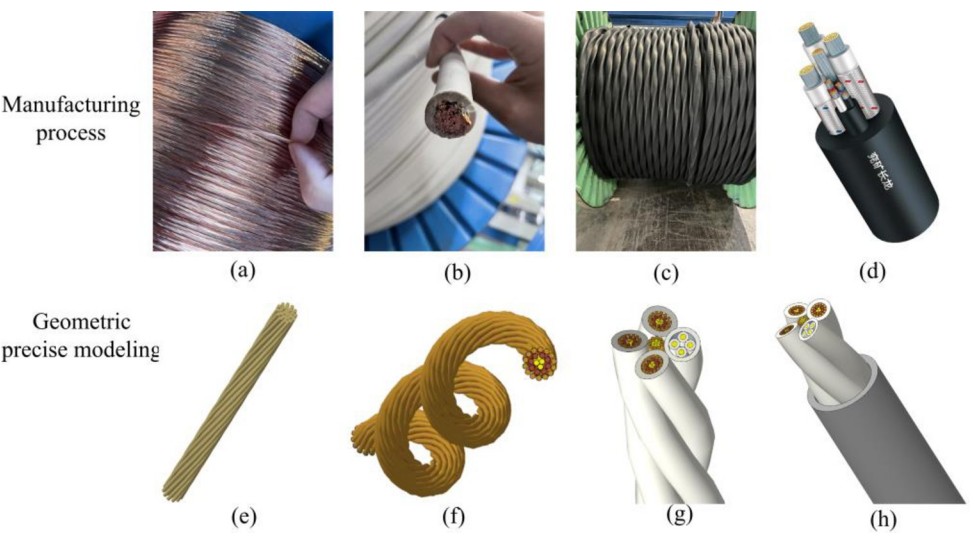

**Fig 3. Comparison of manufacturing process and 3D accurate modeling.**

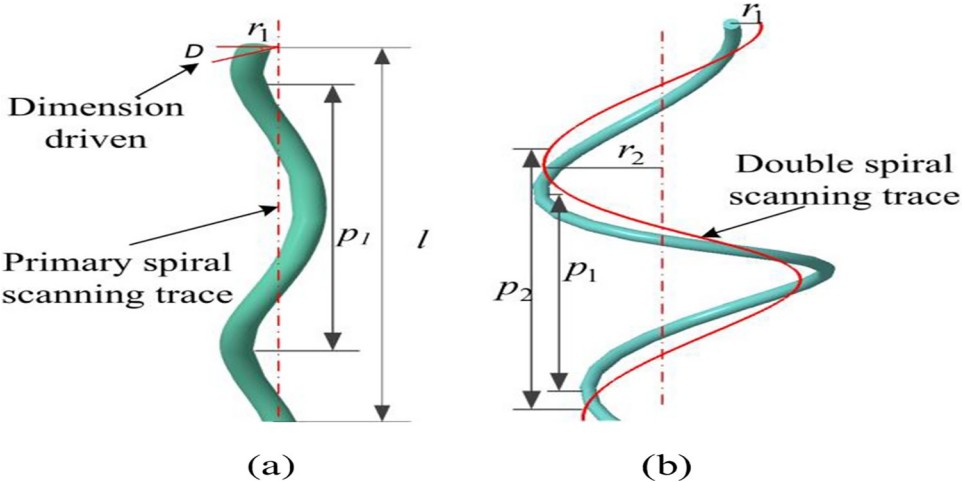

**Fig 4.** Schematic diagram of spiral structure: (a) primary spiral; (b) double spiral.

Fig 4 shows a schematic diagram of the mathematical principles of forming primary and double spiral structures. The lay ratio is the ratio of pitch $p$ to diameter $r$, which determines the mechanical strength of the conductor after stranding. The cable 3D geometry is accurately modeled by using the trajpar function as the dimension driven, with the cross section perpendicular to the spiral scan trajectory. The trajpar function is a variable that varies linearly from 0 to 1, representing the percentage of length relative to the origin trajectory. Define the axial length as $l$, $D$ is the driving dimension, and $t$ is the unit time. The primary spiral structure should satisfy Formula (1), and its spiral scanning trajectory is a straight line. The scanning trajectory of the double spiral structure is a spiral curve, which should satisfy both Formulas (1) and (2):

$$D = 360 trajpar(l/p_1) \tag{1}$$

$$\begin{cases} r = r_2 \\ theta = 360t(l/p_2) \\ z = lt \end{cases} \tag{2}$$

## 3 Calibration of cable equivalent parameters based on tensile test and finite element method

### 3.1 Calibration of the equivalent parameters of the smallest unit strand

The smallest unit of the modeling analysis is the strand composed of many copper wires. By calibrating its equivalent density and equivalent elastic modulus, the complex structure is simplified into an equivalent cylindrical structure with the same outer diameter and mechanical properties, the equivalent principle is shown in Fig 5. Through the ratio of the total cross-sectional area of the copper wires to the cross-sectional area of the equivalent cylinder, the equivalent density of the power and ground strand is calculated to be 6343 kg / m$^3$, and the equivalent density of the control strand is 6041 kg / m$^3$.

The tensile test and the finite element method were combined to calibrate the equivalent elastic modulus of the power and control strands. As shown in Fig 6A, samples of power and control strands with a specimen length of 100 mm were prepared. The test was performed using an MTS CMT5205 tensile testing machine, and the tensile speed was set to 100 mm/

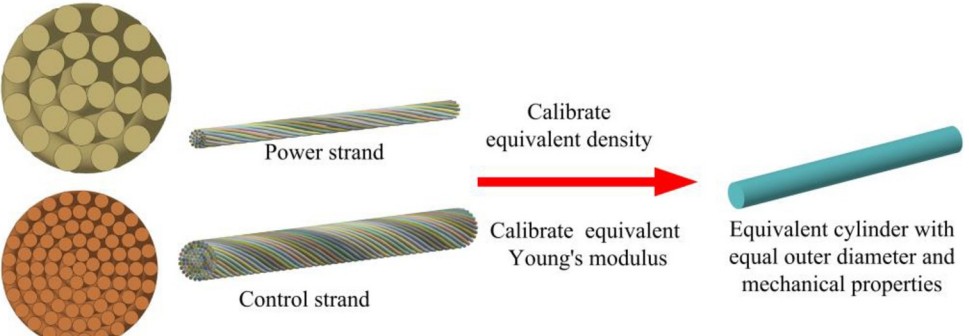

**Fig 5. Equivalent schematic diagram of minimum analysis unit strand.**

min, and the data points were sampled once at 0.032 s. The force and displacement measured by the tensile test were converted into stress and strain, and the stress-strain curves were plotted by Origin and compared with the finite element simulation results, as shown in Fig 6B. The finite element simulation boundary conditions were idealized with fixed constraints, while the relative sliding of the clamp and the sample in the tensile test could produce errors, resulting in the displacement values larger than the real values. Within the elastic deformation range, according to the slope of the stress-strain curve, the equivalent elastic modulus of power strand was calibrated to be 80 GPa and the control strand was 70 GPa.

Within the elastic deformation range, the material parameters for each component unit of the cable are determined as shown in Table 1.

## 3.2 Calibration of the equivalent parameters of orthotropic elastomeric cable

The homogenization theory can not only analyze the equivalent modulus and deformation of materials from the micro scale, but also analyze the stress-strain response of structures from the macro scale [25]. The cable is homogeneous at the macroscopic scale and can be seen as a periodic repetitive stacking of many unit cells in space. As shown in Fig 7, the cable has the same cross-sectional structure and properties. From section A to Section D, the cross-section

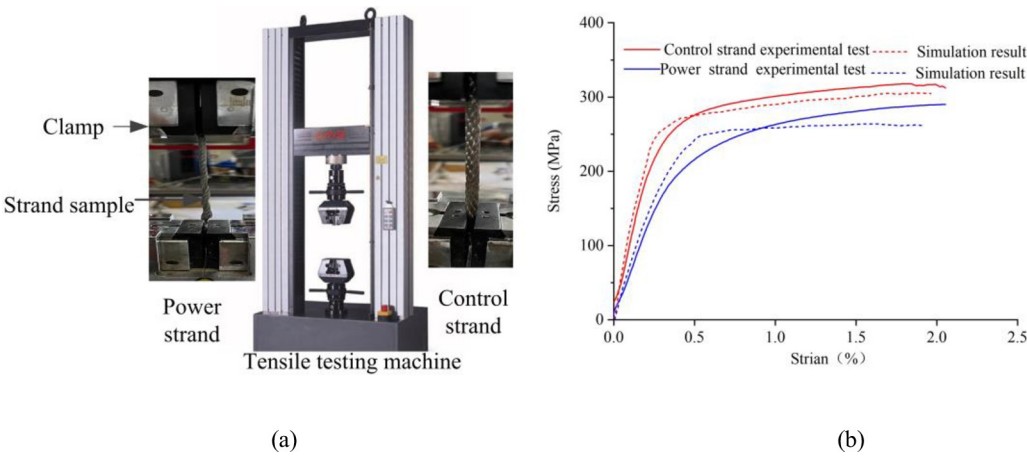

(a)                                                                                              (b)

**Fig 6.** Tensile test: (a) test equipment; (b) stress-strain curve.

**Table 1. Material parameters of each component unit of the shearer cable.**

| Constituent unit | Material | Density (kg/m³) | Equivalent elastic modulus (MPa) | Poisson's ratio | Tensile strength (MPa) |
|---|---|---|---|---|---|
| Power strand | Tinned copper | 6343 | 80000 | 0.34 | 270000 |
| Ground strand | Tinned copper | 6343 | 80000 | 0.34 | 270000 |
| Control strand | Tinned copper | 6041 | 70000 | 0.31 | 250000 |
| Insulation | EPDM rubber | 1500 | 7.8 | 0.47 | 11 |
| Outer sheath | Neoprene | 1250 | 6.11 | 0.47 | 17 |

rotates around the center line once with the period as the pitch. The prediction of equivalent parameters for homogeneous macroscopic structures and non-homogeneous structures with periodic distribution characteristics can be transformed into microscopic homogenization and macroscopic homogenization problems, which can be solved by well-established finite element methods.

The essence of finite element method is to use calculus to subdivide the continuous whole model into several small elements, analyze and accumulate each small element to get the physical properties shown by the whole model. According to the volume average principle, the equivalent physical parameters in a certain direction of the cable can be obtained by the discrete or continuous Formula (3):

$$\begin{cases} \zeta^{eq} = \dfrac{\displaystyle\sum_{i=1}^{N}\zeta_i}{\displaystyle\sum_{i=1}^{N}V_i} \\ \zeta^{eq} = \dfrac{1}{Vol}\displaystyle\int_{V}\zeta\,\partial V \end{cases} \tag{3}$$

Where $Vol$ is the total volume of all elements in the model, $V_i$ is the volume of the $i$-th element, $N$ is the total number of elements, and $\zeta$ is the target physical parameters such as stress or strain.

The cable was assumed to be an orthotropic elastic model with plane xoy, xoz and yoz as its three mutually perpendicular planes of elastic symmetry. The normal equivalent elastic modulus of the cross-section and the equivalent shear modulus of the plane yoz and xoz were calibrated by the finite element method. As shown in Fig 8, the spiral structure of the strands is

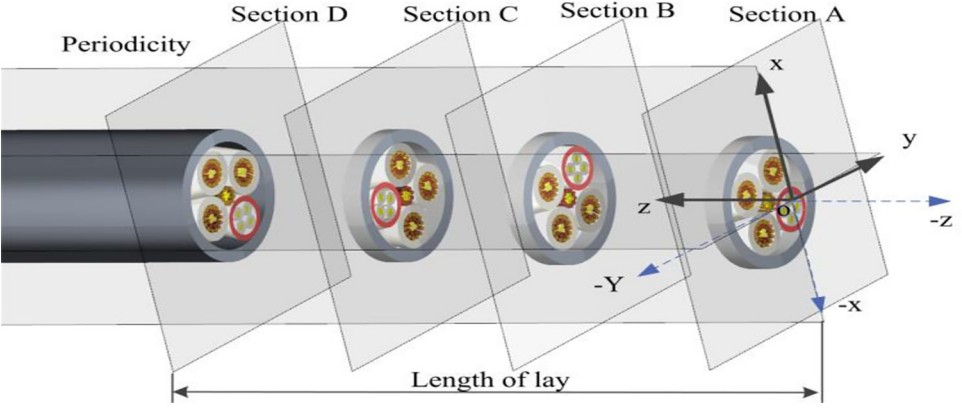

**Fig 7. Schematic diagram of orthotropic elastic model of cable.**

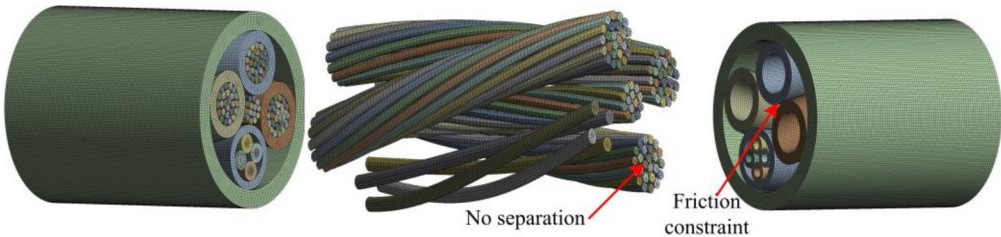

**Fig 8. Finite element model.**

retained. The finite element model of cable was established in Ansys software and divided into high-quality hexahedral meshes with the number of elements of 319861. By defining the contacts between constituent units of the cable, penetration between the units was prevented. In order to speed up the convergence of the solution, the contact type of the strands was defined as non-separation, allowing a small amount of tangential slip, and the contact type with the outer sheath was defined as frictional constraint. The simulation solution was not easy to converge when the geometric large deformation nonlinearity and contact state nonlinearity were considered comprehensively, and the cumulative number of simulation iterations reached 33. To simulate the axial stretching of the cable, the boundary conditions were set to be fixed at one end-section and the other end-section was subjected to the force of 2000 N. To simulate the cable subjected to shearing action, one end-section was also set to be fixed, while the force of 500 N was applied to the other end-section in the directions of X positive, X negative, Y positive and Y negative, respectively.

As shown in Fig 9A, the maximum deformation of the cable in the Z direction is 4.1 mm. After processing the data points of the finite element simulation results, the equivalent average stress-strain curve was plotted and linearly fitted, as shown in Fig 9B. The slope of the linear fitting line is 1.421, which has a good linear fit with the coefficient of determination $R^2$ of 0.996. Therefore, the equivalent elastic modulus of normal to the cross-section was calibrated to be 142 MPa.

Fig 10A–10D show the maximum deformations of the cable corresponding to 6.95 mm, 7.96 mm, 7.71 mm and 7.02 mm in the X positive, X negative, Y positive and Y negative

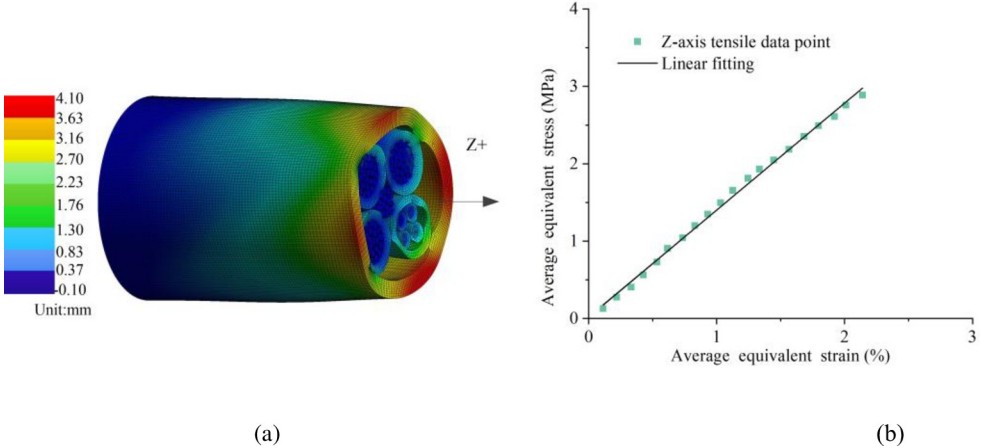

(a)                                                                                    (b)

**Fig 9.** Simulated cable subjected to axial tension: (a) deformation in the Z positive direction; (b) equivalent stress-strain curve.

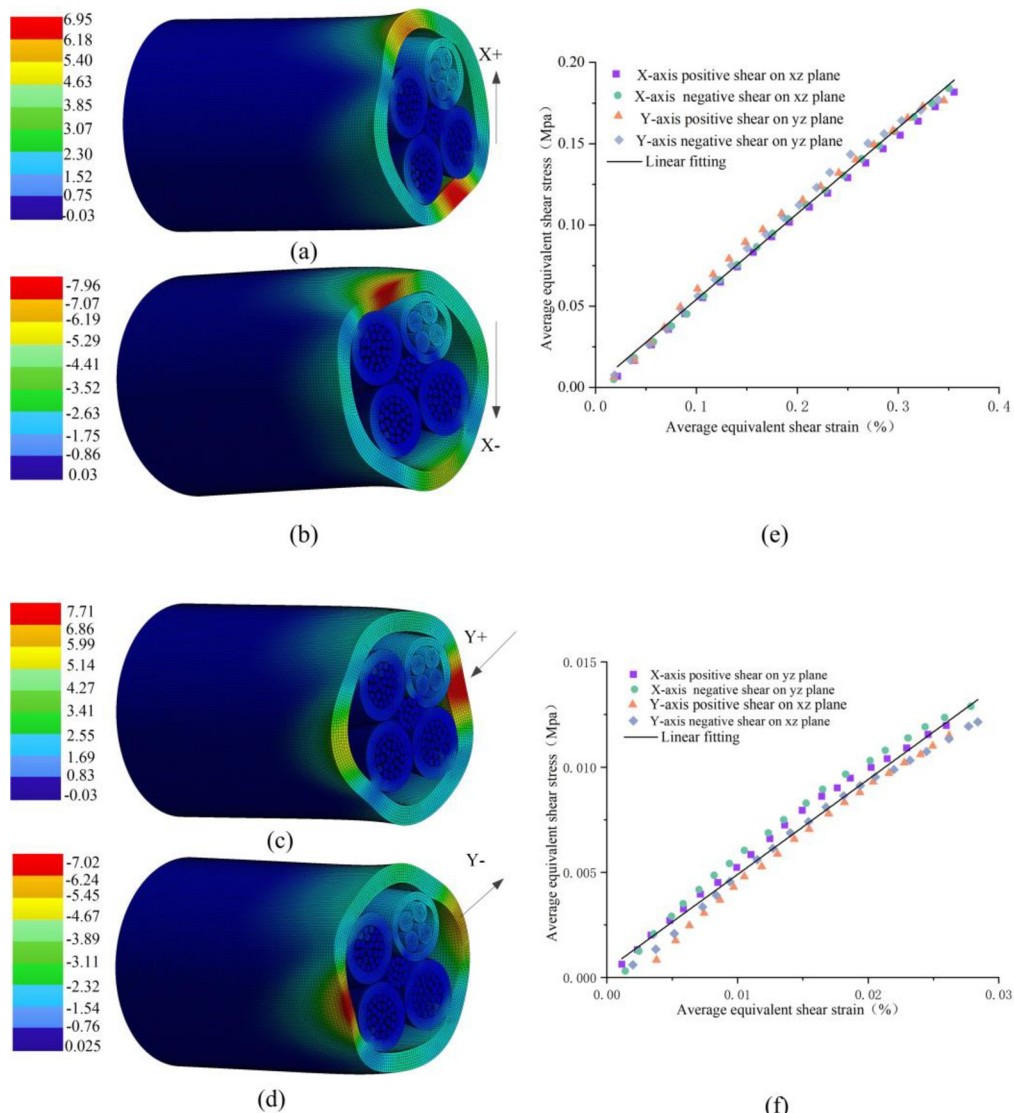

**Fig 10.** Simulated cable subjected to shear: (a) deformation in X positive direction; (b) deformation in X negative direction; (c) deformation in Y positive direction; (d) deformation in Y negative direction; (e) equivalent average shear stress-strain curve of plane yoz; (f) equivalent average shear stress-strain curve of plane xoz.

directions, respectively, reflecting the symmetry of the plane yoz and the plane xoz from the perspective of directional displacement. While the shear strain describes the angle changes in the deformation. As shown in Fig 10E, the simulation data were processed to plot the equivalent average shear stress-strain curve of the plane yoz. The slope of the linear fitting line is 0.528 and the coefficient of determination $R^2$ is 0.993, which has a good linear fit and reflects the symmetry of the plane yoz. So the equivalent shear modulus of plane yoz was calibrated to be 53 MPa. As shown in Fig 10F, the equivalent average shear stress-strain curve of plane xoz was plotted similarly. The slope of the linear fitting line is 0.451 and the coefficient of determination $R^2$ is 0.972, which reflects the symmetry of the plane xoz. So the equivalent shear modulus of plane xoz was calibrated to be 45 MPa.

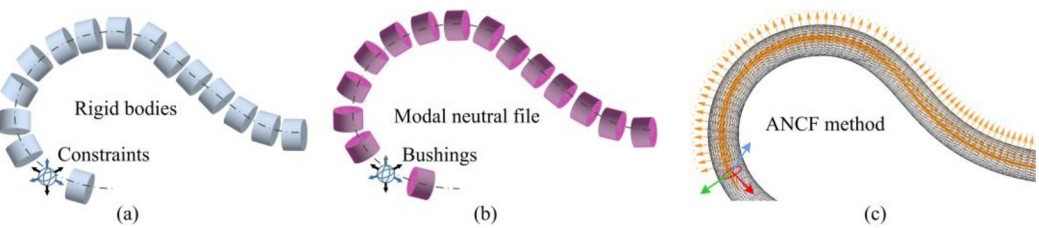

**Fig 11. Schematic and Comparison of different methods about geometric nonlinearity.**

## 4 Dynamic characteristics of shearer cable based on ANCF

### 4.1 Construction of kinetic model of the shearer cable

The main methods for dynamic modeling of cables with large geometrically nonlinear deformations are discrete rigid bodies, multi modal neutral files, and ANCF method. As shown in Fig 11A, a series of rigid bodies are connected by forces to simulate the movement of the cable, but the local deformation cannot be described. As shown in Fig 11B, a series of modal neutral file (MNF) based linear flexible bodies connected by fixed joint type constranints or bushings, which can describe local minor deformation. However, compared to ANCF, the first two discrete methods have certain limitations, such as the physical modeling and the production of neutral files are time-consuming, and the discrete units need to apply contact forces with the cable clamps, resulting in a greater complexity of the simulation analysis.

As shown in Fig 11C, based on the ANCF shear deformable beam theory and the geometrically exact beam formulation, the dynamics model provides a good description of the stretching, shearing, bending and twisting of the cable.

As shown in Fig 12, the coordinates of any node of the cable can be described by its centroid line $r(x,t)$ and cross section moving frame $A(x,t)$:

$$r^p = (x, y, z, t) = r(x, t) + A(x, t)p \qquad (4)$$

Where $p = (x,y,z)^T$ defines the coordinates of node in the cross section reference frame, the transformation matrix $A(x,t) = (a,b,c)$ is described by the rotation vector $\theta(x,t)$. Rodrigues rotation formula is widely used in the field of space analytic geometry and computer graphics,

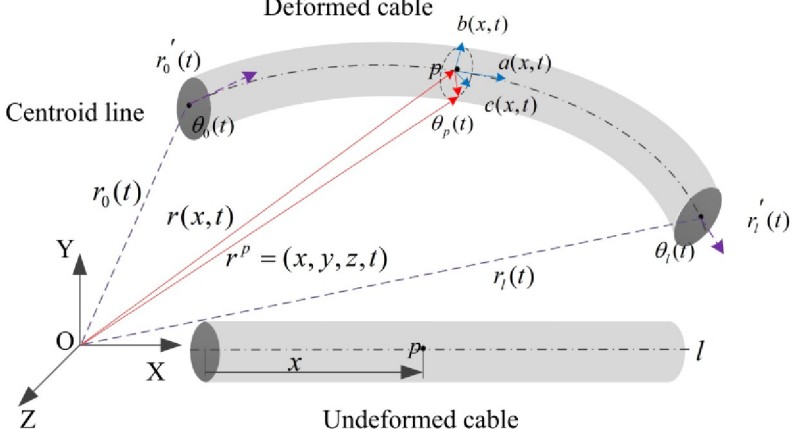

**Fig 12. Geometric description of generalized coordinates and deformation of cable.**

and becomes the basic calculation formula of rigid body motion. So the transformation from rotation vector to rotation matrix is expressed by formula [26]:

$$A = I + \frac{\sin\theta}{\theta}\langle\theta\rangle + \frac{1-\cos\theta}{\theta^2}\langle\theta\rangle^2 \tag{5}$$

Where I is the third order unit matrix, the angle of rotation is represented by the module of $\theta$, $\langle\theta\rangle$ is defined as the antisymmetric matrix, as shown in Formulas (6) and (7):

$$\theta = \|\theta\| = \sqrt{\theta_1{}^2 + \theta_2{}^2 + \theta_3{}^2} \tag{6}$$

$$\langle\theta\rangle = \begin{bmatrix} 0 & -\theta_3 & \theta_2 \\ \theta_3 & 0 & -\theta_1 \\ -\theta_2 & \theta_1 & 0 \end{bmatrix} \tag{7}$$

The length of the cable is $l$, the $x$-coordinates of the points on the cable centroid line range from 0 to $l$. Define $\xi$ as the normalized arc length coordinate in the cable, which is expressed by the ratio of $x$ and $l$. The position vectors $r(x,t)$ and the rotation vectors $\theta(x,t)$ are independently interpolated by cubic Hermite interpolation, as shown in Formula (8):

$$\begin{cases} r(x,t) = (1 - 3\xi^2 + 2\xi^3)r_0(t) + l(\xi - 2\xi^2 + \xi^3)r_0'(t) + (3\xi^2 - 2\xi^3)r_1(t) + l(\xi^3 - \xi^2)r_1'(t) \\ \theta(x,t) = (1 - 3\xi + 2\xi^2)\theta_0(t) + 4\xi(1 - \xi)\theta_p(t) + \xi(2\xi - 2)\theta_1(t) \end{cases} \tag{8}$$

Through the above derivations, the generalized coordinate expression for node p of the cable can be obtained as:

$$q = (r_0{}^T r_0'{}^T \theta_0{}^T \theta_p{}^T r_1{}^T r_1'{}^T \theta_1{}^T)^T \tag{9}$$

From the perspective of energy, the dynamics of the cable system is analyzed based on the Lagrange's equation, and derive the mathematical expressions for kinetic energy, elastic potential energy, and virtual work [27]. By solving the Kinetic equations, the dynamical parameters of any node of the cable can be obtained, including the generalized coordinates consisting of global coordinates and Euler angles, translational and angular velocities, translational and angular accelerations, elastic forces and moments. And then, the stress and strain at each point can be solved from the stiffness matrix. Formula (10) is the Lagrange's equation:

$$\frac{d}{dt}\frac{\partial E^{k0}}{\partial q_i} - \frac{\partial E^{k0}}{\partial q_i} + \frac{\partial E^{p0}}{\partial q_i} + \frac{\partial \psi^T}{\partial q_i}\lambda = Q_i \tag{10}$$

Where $E^{k0}$ is total kinetic energy of cable, $E^{p0}$ is total elastic potential energy of cable, $\Psi$ is the constraint equations, $\lambda$ is the Lagrange multipliers for constraints, and $Q$ is the generalized applied forces. As shown in Formula (11), the kinetic energy of the cable can be expressed as the sum of the translational kinetic energy and the rotational kinetic energy:

$$E^k = \frac{1}{2}\int_0^1 (\rho A\dot{r}^T\dot{r} + \omega^T J\omega)dx \tag{11}$$

Where $\rho$ is density of cable, $A$ is area of cross section, $J$ is rotary inertia of the cross section. $\omega$ is angular velocity of moving frame, which can be derived from Formulas (12) and

(13):

$$\omega = B\dot{\theta} \tag{12}$$

$$B = I - \frac{1 - \cos\theta}{\theta^2}\langle\theta\rangle + \frac{\theta - \sin\theta}{\theta^3}\langle\theta\rangle^2 \tag{13}$$

Elastic potential energy at point $p$ on the cable can be derived from Formula (14):

$$E^p = \frac{1}{2}\int_0^1 [(\gamma^T C_A \gamma) + (\kappa - \kappa_0)^T C_I(\kappa - \kappa_0)]dx \tag{14}$$

Where $\gamma$ is strains of the centroid line, $\kappa$ is curvature of the moving frame, $C_A$ is stiffness matrix of the strain, and $C_I$ is the stiffness matrix of the curvatures. Substitute the equivalent parameters of the previously calibrated cable, the stiffness matrix and the rotational inertia matrix can be derived from Formulas (15) and (16):

$$C = \begin{bmatrix} E_{xx} & 0 & 0 \\ 0 & G_{xy} & 0 \\ 0 & 0 & G_{yz} \end{bmatrix} \tag{15}$$

$$I = \begin{bmatrix} I_{xx} & I_{xy} & I_{xz} \\ I_{yx} & I_{yy} & I_{yz} \\ I_{zx} & I_{zy} & I_{zz} \end{bmatrix} \tag{16}$$

The virtual work done by the external force at point $p$ on the cable can be derived from Formula (17):

$$\delta W^p = \int_0^l [\delta r^T(x,t)F(x,t) + (B^T\delta\theta)^T(x,t)T(x,t)]dx \tag{17}$$

Where $F$ is applied distributed force on the cable, and $T$ is applied distributed torque on the cable. The relationship between force and stretch-shear strain and the relationship between torque and torsion-bending curvatures can be expressed by Formula (18):

$$\begin{cases} \dfrac{F}{A} = C\gamma \\ \dfrac{T}{I} = C\kappa \end{cases} \tag{18}$$

Combining the above derived formulas, then the strain of the cable in the coordinate ($\xi,y,z$) is obtained from Formula (19), and the corresponding stress is obtained from Formula (20):

$$\begin{cases} \varepsilon_{xx}(\xi,y,z) = \gamma_1(\xi) - y\kappa_3(\xi) + z\kappa_2(\xi) \\ \gamma_{xy}(\xi,y,z) = \gamma_2(\xi) - z\kappa_1(\xi) \\ \gamma_{xz}(\xi,y,z) = \gamma_3(\xi) - y\kappa_1(\xi) \end{cases} \tag{19}$$

$$(\sigma_{xx}, \tau_{xy}, \tau_{xz})^T = C(\varepsilon_{xx}, \gamma_{xy}, \gamma_{yz})^T \tag{20}$$

## 4.2 Analysis of dynamic characteristics of shearer cables for different working faces and towing systems

The current testing of the mechanical properties of the cable before leaving the factory includes mechanical shock resistance test, extrusion resistance test and bending resistance test. Fig 13A shows the shearer cable bending testing machine of Shandong Energy Group. By setting the moving speed of the upper plate to simulate the reciprocating motion of the shearer and using a simple arch plate to simulate the complex geological conditions of the working face, the lower end was fixedly connected to the power supply. The bending times of the cable when broken or short circuited were taken as the evaluation basis for the life of the shearer cable. This test has certain limitations, as it fails to take the coal seam conditions, coal mining process, kinematics and dynamics of the shearer into account, so that it cannot accurately reflect the actual working dynamic load of the cable. As shown in Fig 13B, an industrial test was carried out at the 2101 working face of Taiyue coal industry. The working face was equipped with 103 scraper conveyors, and tension sensors were arranged at the connection between the cable traction device of the shearer and the cable clamps to indirectly monitor the tension on the cables, and the measured data of the maximum stroke full load tension of the cable traction clamps of the coal miner were less than or equal to 3000 N.

Based on the ANCF method, a rigid-flexible coupled virtual prototype co-simulation model of the cable towing system of shearer was established to directly obtain the dynamic characteristics parameters of the cable. The new intelligent cable towing system is shown in Fig 14A, which can realize the cable towing device adaptively following the operation of shearer and improve the intelligence and unmanned operation of fully mechanized mining face. The round-link chain drive system pulls the towing trolley to follow the shearer to do reciprocating movement. The shearer cable is protected in the clamps and only bends once at the towing trolley. By controlling the speed of the towing trolley to be half of the speed of shearer, it makes the whole constitute a movable pulley system [28, 29]. Besides, the hydraulic tensioning system is arranged at the driven sprocket to solve the problem of falling of chains under gravity

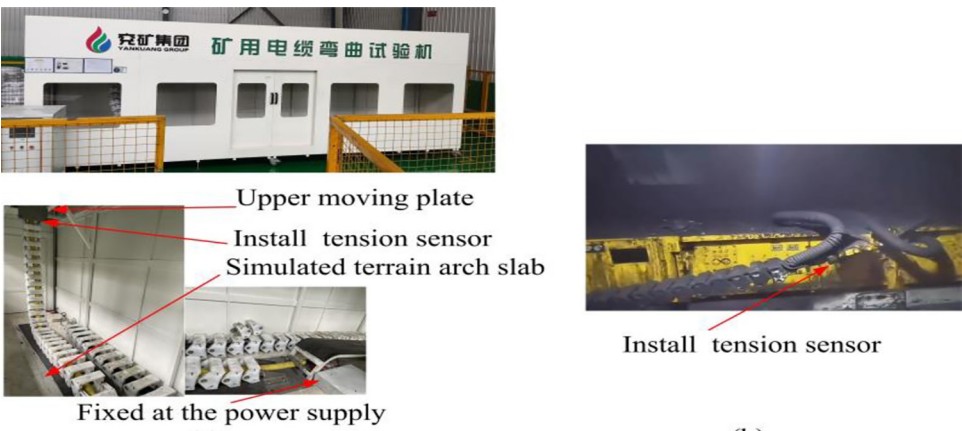

**Fig 13.** Indirect acquisition of cable tension by test methods: (a) cable bending test; (b) industrial test.

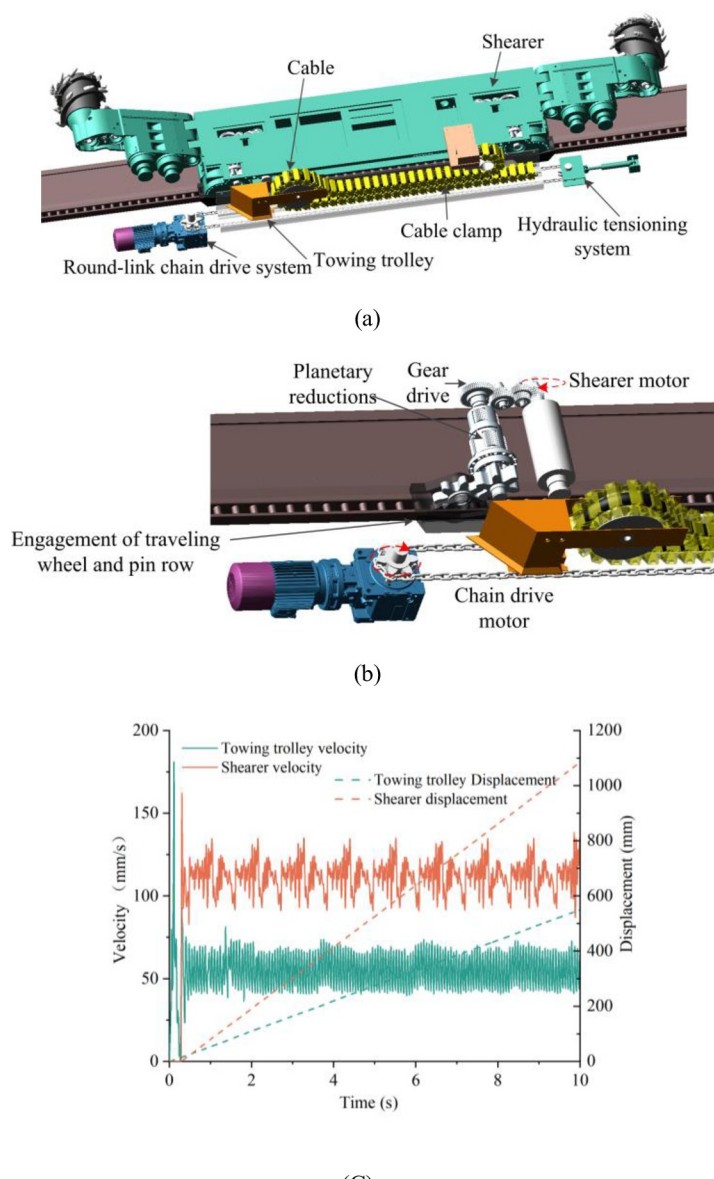

**Fig 14.** Rigid-flexible coupling model of the intelligent towing system: (a) overall system diagram; (b) internal transmission diagram; (c) speed and displacement curves of towing trolley and shearer.

by applying preload. Fig 14B shows a schematic diagram of the internal drive of the whole system. Considering the internal gears drive of the shearer, the planetary wheel deceleration and the engagement of the walking wheels and the pin row, and the fluctuation characteristics of the long-distance transmission of the round-link chains, the speed and displacement curves of the towing trolley and the shearer are shown in Fig 14C.

The Fe-part model of the cable based on the ANCF method in was constructed in Adams software, as shown in Fig 15. The centroid line of the cable is a B-spline curve formed by mass center of the cable clamps. By setting appropriate intermediate nodes can both improve the accuracy of the cable model and speed up the simulation, 400 intermediate nodes are inserted in this simulation. Define the cross-sectional parameters, equivalent damping ratio and

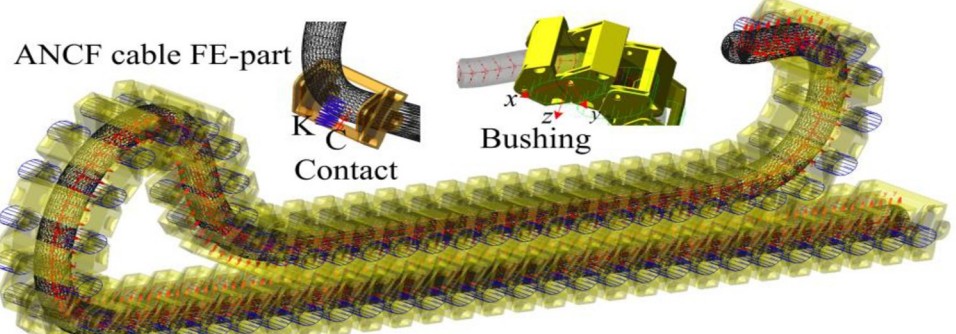

**Fig 15. Fe-part model of shearer cable based on ANCF method.**

equivalent parameters of orthotropic elastomeric cable. As shown in Formula (21), compared to conventional rotational constraints, the bushing connections between cable clamps allow for a small amount of displacement in the lateral and longitudinal directions, in addition to mutual rotation [30]. The way in which the bushings are applied matches the actual state of forces on the pins between the cable clamps, while the allowance of restraint speeds up the convergence of the kinetic equations. The method of macro command flow is used to define the bushing connections between two adjacent cable clamps, the possible contact force between clamps and the contact forces between clamps and the cable. The contact force parameters are determined the material, $K$ is the stiffness coefficient, and $C$ is the damping coefficient. By this way, forces, moments, stresses, strains, bushings and contact forces at each node of the cable can reflect the tension state and dynamic gradual change characteristics of the cable from multiple perspectives.

$$\begin{bmatrix} F_x \\ F_y \\ F_z \\ T_x \\ T_y \\ T_z \end{bmatrix} = -K \begin{bmatrix} x \\ y \\ z \\ a \\ b \\ c \end{bmatrix} - C \begin{bmatrix} V_x \\ V_y \\ V_z \\ \omega_x \\ \omega_y \\ \omega_z \end{bmatrix} + \begin{bmatrix} F_1 \\ F_2 \\ F_3 \\ T_1 \\ T_2 \\ T_3 \end{bmatrix} \tag{21}$$

Adjacent cable clamps are subject to torsion and shear in $x$ and $y$ direction, and subject to tension in $z$ direction. Where $F$ and $T$ are force and moment, $x$, $y$, $z$ are relative displacements, $a$, $b$, $c$ are angles, $V_x$, $V_y$, $V_z$ are relative velocities, $\omega_x$, $\omega_y$, $\omega_z$ are relative angular velocities, $F_1$, $F_2$, $F_3$, $T_1$, $T_2$, $T_3$ are preloads in six directions, $K$ is the diagonal matrix of stiffness coefficients, $C$ is the diagonal matrix of damping coefficients.

The model of MG2×70/325-BWD thin coal seam shearer is used for the simulation. Its mining height ranges from 0.8 m to 1.55 m, maximum towing speed can research 6.86 m/min, and its working face inclination is less than or equal to 35 degrees. The shearer cable towing tension is basically proportional to the length of the working face and has a good linear relationship. Based on the similarity criterion, the ratio of the cable tension obtained from this simulation to the actual working surface is 1: 2.5. For the dynamics simulation of different working surface inclination and different towing systems, the specific working condition parameters are set in Table 2, and Fig 16 shows the simulation of four working conditions. The

**Table 2. Parameters of four different working conditions.**

| Simulation | Towing way | Towing velocity (m/min) | Inclination of working face (°)s | Direction of advance |
|---|---|---|---|---|
| 1 | Intelligent towing | 6 | 0 | Shearer in the front |
| 2 | Intelligent towing | 6 | 30 | Towing upward with shearer in the front |
| 3 | Intelligent towing | 6 | 30 | Towing downward with trolley in the front |
| 4 | Traditional Towing | 6 | 0 | Shearer in the front |

cable is divided into four working areas according to its dynamic gradient characteristics and movement trends. As shown in Fig 16A, area (a) is near the shearer traction device, area (b) is positioned in the upper part of the bend of the towing trolley while area (c) is positioned in the lower part, and area (d) is near the fixation of the power supply. As shown in Fig 16A, the

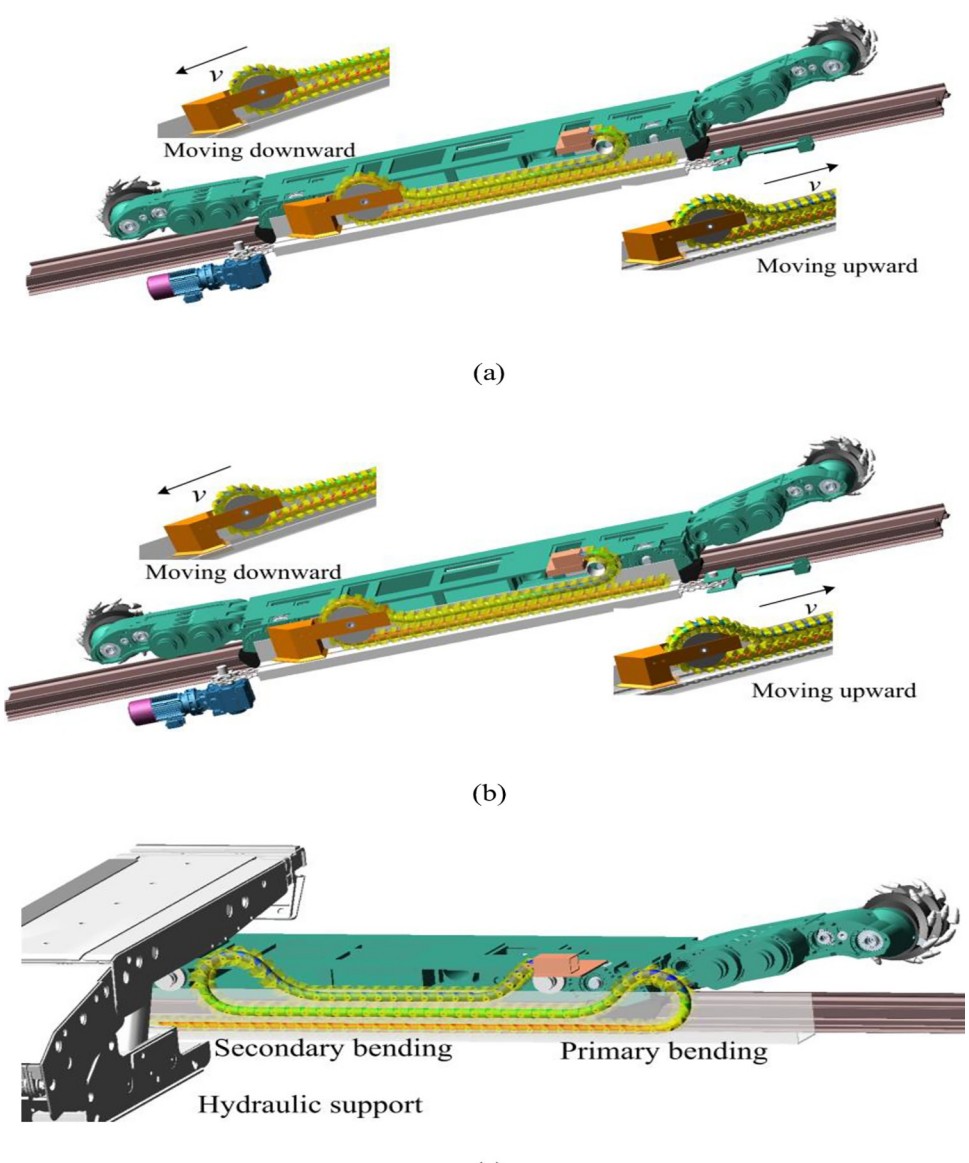

**Fig 16.** Dynamics simulations: (a) intelligent towing at horizontal working surface; (b) towing upward and downward at large inclination; (c) conventional towing at horizontal working surface.

bushing 3 is located in area (a), and the contact force 35 is located in area (c). The kinetic and kinematic parameters of Node54, Node199, Node220 and Node370 in different corresponding areas are extracted, and the simulation result data are plotted shown in Fig 18.

The simulation results are analyzed as follows:

(1) By dynamically monitoring bushings and contact forces, it is possible to reflect the tension state and dynamics characteristics of the cable at that location. As shown in Fig 17A, the three-dimensional force curves of the contact force 35 reflects the contact behavior of the cable at the bend of the towing trolley. At the moment of 1.5 s, the cable clamp and the cable start to contact, the cable is bent by the shear force of the cable clamp in X and Y directions, the contact force increases and reaches the maximum value at 4 s. The cable and the 35th cable clamp are gradually separated, and then, the contact force disappears after complete separation. The cable Fe-part model is to subdivide a whole cable into many continuous segments,which is very similar to the overall structure of the cable clamps system. As shown in Fig 17B, the bushing 3 is located near the traction of the shearer, so its tension state is similar to that of the cable in area (a). The pined connections between adjacent cable clamps are simulated by two bushings, and the amplitude of bushing 3 in this simulation is 500 N. According to the similarity criterion, the actual bushings are calculated to be 2500 N, which is consistent with result that the measured data of the maximum full load tension of the cable clamp in the industrial test is less than or equal to 3000 N. This also proves the rationality of this method to obtain the parameters of the dynamic characteristics of the cable. The material stiffness of both the rubber sheath of the cable and the nylon of the cable clamp are not large. If an unusually high contact force is monitored, large gangue could be present here, posing a threat to the reliability of the cable.

(2) Unlike bending tests and industrial tests where the tension between the traction device and the cable clamp were measured indirectly to reflect the force on the cable, the ANCF cable model can be used to directly extract the kinetic and kinematic parameters at each node of the cable in the towing process. The cable in area (a) is close to the shearer traction device and is stretched by the shearer in the x-direction. As shown in Fig 18A, the component Fx of Node 54 increases with the towing mass of the cable and the cable clamp gradually becoming larger. At the moment of 6 s, the cable and clamps are in tension equilibrium,

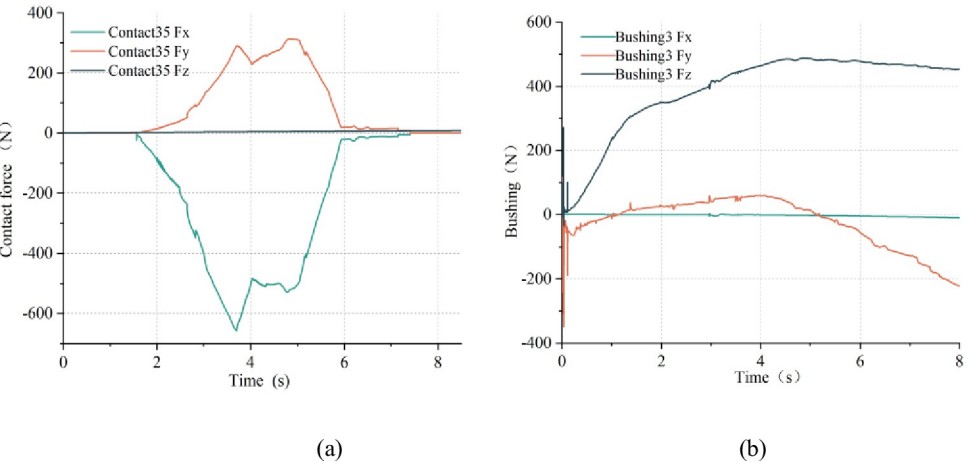

(a) (b)

**Fig 17.** Curves of contact force and bushing: (a) contact force 35; (b) bushing 3.

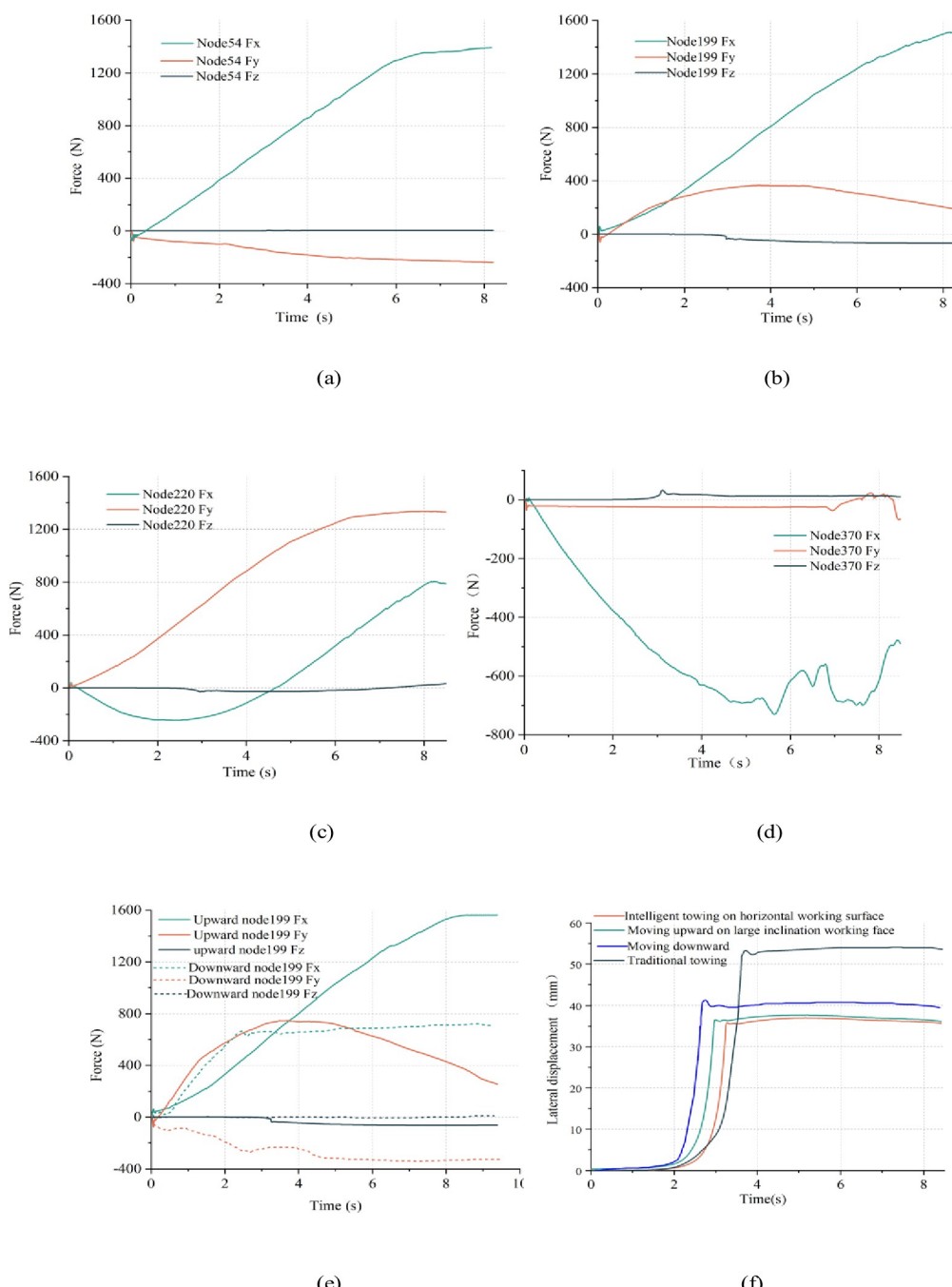

**Fig 18.** Dynamic gradient changes of the cable: (a) three-dimensional force in area a; (b) three-dimensional force in area b; (c) three-dimensional force in area c; (d) three-dimensional force in area d; (e) comparison of towing upward and downward at large inclination; (f) comparison of lateral displacement with different towing systems.

while the component Fx will not increase and remain at 1300 N. As shown in Fig 18B, the cable in area (b) is located at the upper half of the bend of the towing trolley and makes bending motion around the roller. The shear component Fy of Node 199 increases and then decreases. Its movement tendency is to transition from bending state of the upper half to the horizontal state. As shown in Fig 18C, Node220 in the lower half of the bend is

subjected to a larger y-direction shear force and is stabilized at 1250 N, while the tension component Fx shows a law of decreasing after increasing in the negative direction and then increasing in the positive direction. As shown in Fig 18D, area (d) is near the fixation of the power supply. As the cable clamps and cables are gradually tensioned, the Node 370 is mainly subjected to tension in the X negative direction, with a maximum value of 700 N.

By analyzing the dynamic gradient characteristics of the cable in four areas, it can be found that the cable in area (a) and (d) are mainly subjected to tensile force in X direction, while area (b) and (c) are mainly subjected to shear force in Y direction. For the optimized design of the shearer cable, the cable should have both good tensile strength and bending strength. For the arrangement of cables in the working face, a certain amount of movement allowance shall be left at the connection with the shearer and the power fixed position to avoid the cable cores being broken by the large axial tension in these two areas.

(3) The towing system at large inclination working face is analyzed from the perspective of simulation and theoretical analysis. According to the classical inclined slider physical model, the towing force upward and downward at large inclination working face can be derived from Formulas (22) and (23):

$$f_{up} = m(t)(a + g\sin\theta + g\mu\cos\theta) \tag{22}$$

$$f_{down} = m(t)(a - g\sin\theta + g\mu\cos\theta) \tag{23}$$

Where $m(t)$ is the towing mass per unit time, including the mass of the cable clamps and cable, $g$ is the gravitational acceleration, a is the acceleration of shearer, $\theta$ is the working face inclination, $\mu$ is the coefficient of dynamic friction of the cable clamp and the cable as a whole with the cable trough. As shown in Fig 18E, the three-dimensional force change trend of the Node220 at the high inclination working face is consistent with that of the horizontal working face, but the moment when the cable and the cable clamps reach the tension balance is delayed and the force is greater than 1500 N. It is necessary to prevent the cable cores from being pulled off due to the large tension during towing upward. However, due to the gravity component of the cable and cable clamps, the required towing force is less than 600 N when towing downward.

(4) As shown in Fig 16C, due to the lack of restrictions of towing trolley on the attitude of the cable, the bending radius of the cable will change uncontrollably during the towing process. The cable moves back and forth with shearer for many times in short distances, forming the state of secondary bending. The cable clamps are stacked too high in the vertical direction, so that there is also a risk of interference with the hydraulic support in thin coal seam working faces. As shown in Fig 18F, the maximum lateral displacement of the conventional towing is 54 mm, followed by 40 mm for towing downward at the large inclination working face, both of which have the risk of slipping out of the cable trough.

## 5 Conclusion

Based on the ANCF method, the rigid-flexible coupled virtual prototype co-simulation model of shearer cable towing system was constructed to obtain the kinetic and kinematic parameters of each node of the cable and study the dynamic gradual change characteristics of the cable in different working areas. This research method has an important theoretical significance and

engineering application value for the acquisition of dynamic characteristic parameters of shearer cables and the optimal design and dynamic reliability of cables. The following conclusions can be drawn:

1. The equivalent parameters of orthotropic elastomer calibrated by the combination of tensile test and finite element method can accurately reflect the mechanical characteristics of large deformation mobile cable, and can describe the stretching, shearing, bending and twisting of the cable, which provides some experience for the equivalent equivalent prediction of other large deformation flexible bodies.

2. The cable Fe-part established based on the ANCF method can deal with the problem of non-linear large deformation of shearer cables at different working face inclination and different towing systems. Through the analysis of the dynamic gradient characteristics of the cables in different areas. For the optimized design of the shearer cable, it is proposed that the cable should have both good tensile strength and bending strength. For the arrangement of cables in the working face, a certain amount of movement allowance shall be left at the connection with the shearer and the power fixed position to avoid the cable cores being broken by the large axial tension in these two areas.

3. Compared with the traditional towing method, intelligent towing method due to the constraints of the towing trolley, the cable will only produce one bend at the towing trolley, which avoids the risk of interference with the hydraulic support caused by secondary bending and the risk of sliding out of the cable tray due to large lateral displacement. This research method of putting the subject cables into different towing systems can provide the reference basis for the design of intelligent and adaptive towing systems.

## Acknowledgments

The authors would like to acknowledge the support and contribution from the State Key Lab of Mining Machinery Engineering of Coal Industry, Liaoning Technical University, China. The authors would also like to thank Changlong cable factory of Shandong energy group for providing cable parameters and experimental conditions.

## Author Contributions

**Formal analysis:** Haining Zhang.

**Investigation:** Liguo Han.

**Methodology:** Lijuan Zhao.

**Resources:** Man Ge.

**Supervision:** Feng Gao.

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
