## [Decision Letter · Decision Letter 0]

6 Dec 2022

PONE-D-22-31523Research on Dynamic Characteristics of Large Deformation Shearer Cable Based on Absolute Node Coordinate FormulationPLOS ONE

Dear Dr. Zhang,

Thank you for submitting your manuscript to PLOS ONE. After careful consideration, we feel that it has merit but does not fully meet PLOS ONE’s publication criteria as it currently stands. Therefore, we invite you to submit a revised version of the manuscript that addresses the points raised during the review process.

We look forward to receiving your revised manuscript.

Kind regards,

Zilin Gao, Ph.D

Academic Editor

PLOS ONE

“The authors would like to acknowledge the support and contribution from the State Key Lab of Mining Machinery Engineering of Coal Industry, Liaoning Technical University, China. The authors would also like to thank Changlong cable factory of Shandong energy group for providing cable parameters and experimental conditions. This work was supported by the National Natural Science Foundation of China [Grant number 51674134], the Liaoning Provincial Natural Science Foundation of China [Grant number 20170540420], and the Key projects of Liaoning Provincial Department of Education [Grant number LJ2017ZL001].”

“This work was supported by the National Natural Science Foundation of China [Grant number 51674134], the Liaoning Provincial Natural Science Foundation of China [Grant number 20170540420], and the Key projects of Liaoning Provincial Department of Education [Grant number LJ2017ZL001].”

“Competing interests

The authors declare no competing interests.”

Reviewers' comments:

Reviewer's Responses to Questions

**Comments to the Author**

1. Is the manuscript technically sound, and do the data support the conclusions?

Reviewer #1: Yes

Reviewer #2: Yes

Reviewer #3: Yes

2. Has the statistical analysis been performed appropriately and rigorously? 

Reviewer #1: Yes

Reviewer #2: Yes

Reviewer #3: Yes

3. Have the authors made all data underlying the findings in their manuscript fully available?

Reviewer #1: Yes

Reviewer #2: Yes

Reviewer #3: Yes

4. Is the manuscript presented in an intelligible fashion and written in standard English?

Reviewer #1: Yes

Reviewer #2: Yes

Reviewer #3: Yes

5. Review Comments to the Author

Reviewer #1: The manuscript entitled ‘Research on Dynamic Characteristics of Large Deformation Shearer Cable Based on Absolute Node Coordinate Formulation’ applies ANCF（Absolute Node Coordinate Formulation） method to study the dynamic characteristics of cable with large deformation. The topic is interesting and needs to be investigated. For the cables of composite materials and complex structural bodies, the equivalent mechanical parameters are reasonably calibrated by finite element method and tensile test method. A rigid-flexible coupling virtual prototype co-simulation model of the cable dragging system of coal mining machine was established to study the dynamic gradual changes of the cable in different working areas to solve the problem that the actual stress state of the cable in the mine is not easy to obtain, which has some theoretical significance and engineering application value. However，some of the following issues need to be discussed and modified for its improvement.

1.The introduction and chapter 4 mentioned the inability of the conventional finite element method to explain the large deformation motion of the cable, which needs to be briefly explained here in comparison with the current methods for studying such large deformation multi-flexible body dynamics.

2.The names of the industrial simulation analysis software used in the text should be reflected in the text.

3.The latest references need to be supplemented.

4.The English language expression can accurately use the proper nouns of the discipline, but there are also some small errors in word spelling and grammar, please check the context carefully.

5.Please read the PLOS ONE journal's instructions for formatting papers carefully, such as title and abstract， and revise them as required.

Reviewer #2: The research method is innovative and explores the solution of large deformation multi-flexible body dynamics problems in engineering practice. Coal mine explosion-proof requirements, complex occurrence conditions, and severe loads make it very difficult to do experiments on site. The manuscript used the ANCF （Absolute Node Coordinate Formulation）method to simulate the real movement and force state of the cable in the coal mine, which has certain advantages. At the same time, the analysis method of geometrically accurate modeling of the cable and the dynamic characteristic laws of each area of the cable have some guidance for the optimal design of the cable and the arrangement of the coal mining machine cable at the working face. But there are some problems that need to be discussed and modified.

1.There are some minor errors in the format of the references.

2.The expression for cable area delineation on page 14 of the main text should be consistent with that in Figure 15, and the same should be consistent with it in the analysis. For example Area (a).

3. The overall English language expression is fluent and accurate, but there are some small errors in the text, such as on page 14,“ is” repeatedly expressed, and on page 17, “curve ”should be used in the plural form, suggesting that the author should check carefully the spelling and grammatical errors.

4.In the chapter 3.2 of equivalent calibration of cable , for the finite element simulation of the cable, whether the spiral structure of the conductor is retained or the equivalent simplification, please show the specific.

5.In the lower label of Figure 17, the initial letters of the subheadings in (e) and (f) should be kept in line with the previous ones and only lowercase.

Reviewer #3: The object of study in this paper is the dynamic characteristics of the mobile rubber-sheathed flexible cables for coal mines, subjected to complex and severe loads. Unlike the static load of laying cable and the periodic load of submarine cable, its deformation will be large when following the coal mining machine for many short round trips.This manuscript uses the ANCF（Absolute Node Coordinate Formulation） method to better model and analyze the dynamics of the cable with large deformation and realize the simulation of simulated movement.The composite material and complex structure of the cable body make it more reasonable to use a combination of experimental, theoretical derivation and simulation analysis research methods.The research in this paper is useful for the simplification of large-deformation multi-flexible systems and complex structural bodies, and is innovative to a certain extent. However, there are also some problems that need to be discussed and modified:

1.The abbreviation of "Discrete Element Method Multi-Flexible Body Dynamics" in the last paragraph of page 3 should be changed from "EDM-FMBD" to "DEM-FMBD". The manuscript should ensure that the proper nouns and abbreviations of the scientific and technical paper are correct, and it is recommended to double-check other proper nouns.

2.The English language expression is overall more fluent, but there are some minor grammatical errors. On page 16, in analysis (1), the sentence "The cable and the 35th cable clamp..." is not clear enough. On page 17, in analysis (3), the word "perspective" should be used in its adverbial form.

3.In chapter 4.1, the "rodrigues formula" is mentioned in equation 5 for the derivation of the dynamics modeling process of large deformation cables.Please give a brief description of its application.

4.On page 13, why bushings are used between adjacent cable clamps rather than revolute joints? Please briefly discuss the advantages of this choice as well as the rationality.

5.It is recommended to add some updated references from year 2022 and revise the format with reference to the journal’s requirements.

6. PLOS authors have the option to publish the peer review history of their article (what does this mean?). If published, this will include your full peer review and any attached files.

Reviewer #1: No

Reviewer #2: No

Reviewer #3: No

---

## [Author Response · Author response to Decision Letter 0]

12 Dec 2022

Dear Editor and Reviewers:

On behalf of my co-authors, we are grateful to you for giving us an opportunity to revise our manuscript. We appreciate you very much for your positive and constructive comments and suggestions on our manuscript entitled “Research on dynamic characteristics of large deformation shearer cable based on absolute node coordinate formulation method”([PONE-D-22-31523]-[EMID：c4f07fcdcbe5401c]).

We have studied reviewers’ comments carefully and tried our best to revise our manuscript according to the comments. The following are the responses and revisions I have made in response to the reviews’ questions and suggestions on an item-by-item basis. Thanks again to the hard work of editor and reviewer!

Response to the comments of Editor

Comment No.1：Please ensure that your manuscript meets PLOS ONE's style requirements, including those for file naming. 

Response: Based on the requirements of the journal formatting requirements document you shared, we have revised the format of the manuscript to ensure that it meets the requirements for journal publication, including headings, references, etc.

Comment No.2：We note that the grant information you provided in the ‘Funding Information’ and ‘Financial Disclosure’ sections do not match.When you resubmit, please ensure that you provide the correct grant numbers for the awards you received for your study in the ‘Funding Information’ section.

Response: 

After careful consideration, and in accordance with your suggestions and journal requirements. We have changed the description of the Financial Disclosure as follows: "This work was supported by the National Natural Science Foundation of China [Grant number 51674134]. ", this section will also be added to the cover letter.

Comment No.3：We note that you have provided funding information that is not currently declared in your Funding Statement. However, funding information should not appear in the Acknowledgments section or other areas of your manuscript. We will only publish funding information present in the Funding Statement section of the online submission form.Please include your amended statements within your cover letter; we will change the online submission form on your behalf.

Response: According to the journal's requirements, 'Funding Information' should not appear in of the text, so we removed that section from the text and added 'Funding Information' to the revised cover letter.

Comment No.4：Please complete your Competing Interests on the online submission form to state any Competing Interests. If you have no competing interests, please state "The authors have declared that no competing interests exist."This information should be included in your cover letter; we will change the online submission form on your behalf.

Response: We have no competing interests. The state "The authors have declared that no competing interests exist."This information should be included in your cover letter.

Comment No.5： In your Data Availability statement, you have not specified where the minimal data set underlying the results described in your manuscript can be found. PLOS defines a study's minimal data set as the underlying data used to reach the conclusions drawn in the manuscript and any additional data required to replicate the reported study findings in their entirety. All PLOS journals require that the minimal data set be made fully available. Upon re-submitting your revised manuscript, please upload your study’s minimal underlying data set as either Supporting Information files or to a stable, public repository and include the relevant URLs, DOIs, or accession numbers within your revised cover letter.

Response: The tensile test data of the minimum unit power strand and control strand for modeling and analysis are uploaded to the public database,and provide a link to the data in the cover letter.

Response to the comments of Reviewer #1

Comment No.1:The introduction and chapter 4 mentioned the inability of the conventional finite element method to explain the large deformation motion of the cable, which needs to be briefly explained here in comparison with the current methods for studying such large deformation multi-flexible body dynamics.

Response:According to your suggestion, at the beginning of chapter 4.1 on page 10 of the manuscript, three diagrams describing the dynamic modeling of geometric nonlinear large deformation, namely discrete rigid bodies, multi modal neutral files, and ANCF method, are added, as shown in Figure 11. The limitations of the first two methods are pointed out from the perspectives of modeling time consumption, whether local small deformation can be described and the complexity of simulation solution. The added text description is also marked in red color.

Comment No.2:The names of the industrial simulation analysis software used in the text should be reflected in the text.

Respone:The finite element software used for the calibration of the equivalent mechanical parameters of the cable body is Ansys, which has been added to the text in section 3.2, page 8, and is marked in red color. The ANCF model of the cable was created in Adams software and it is added in the next paragraph of Figure 15 on page 14 and marked in red.

Comment No.3:The latest references need to be supplemented.

Respone：The literature on prefaces to research in related fields in recent years has been carefully read, and the discussion in the first paragraph of the introduction has been supplemented by the literature on large deformation polymorphic studies [7], [8] and [9], and in the second paragraph of the introduction by the literature on cable equivalence analysis studies [14] and [15], and the manuscript has been added to the end of the Reference with these five related literature in the preamble of the field. The changes have been marked in red color.

Comment No.4：The English language expression can accurately use the proper nouns of the discipline, but there are also some small errors in word spelling and grammar, please check the context carefully.

Respone: The language spelling and grammar of the article has been carefully checked and corrected according to your suggestions. The parts that have been revised have been marked in red color.

Comment No.5：Please read the PLOS ONE journal’s instructions for formatting papers carefully, such as title and abstract, and revise them as required.

Respone: The PLOS ONE journal's instructions for formatting papers have been read carefully and, as requested, the title of the manuscript has been revised so that only the first letter of the title is capitalized and the rest of the words, except for proper nouns, are in lowercase form. "Abstract" is written in the same format as the first level title. Also, the citation format for references has been changed, with "[1]" replacing the previous form of numbers appearing in the upper corner. In accordance with the standard, the DOI link to the reference has been added to the reference section. The descriptive information for 'Competing interests' and 'Acknowledgments' has been removed from the text in accordance with journal formatting requirements and added to the revised cover The letter has been added to the revised cover letter.All the parts mentioned above that have been changed are marked in red.

Response to the comments of Reviewer #2

Comment No.1：There are some minor errors in the format of the references.

Response: Carefully check the journal’s requirements for the format of references and make the following changes：“Abstract” is written in the same format as the first level title. Also, the citation format for references has been changed, with “[1]” replacing the previous form of numbers appearing in the upper corner. In accordance with the standard, the DOI link to the reference has been added to the reference section.The references to the Master's and Doctor's theses are marked as “M.Sc. Thesis” and “Ph.D. Thesis”. All the parts mentioned above that have been changed are marked in red in the context and reference section.

Comment No.2：The expression for cable area delineation on page 14 of the main text should be consistent with that in Figure 15, and the same should be consistent with it in the analysis. For example Area (a).

Response:The description of the division of areas on page 14 has been aligned with that in Figure 16 and revised to read "Area (a)", and the descriptions of other figures and corresponding paragraphs throughout the text have been carefully checked to ensure the rigour of the manuscript. The changes have been marked in red.

Comment No.3：The overall English language expression is fluent and accurate, but there are some small errors in the text, such as on page 14,“ is” repeatedly expressed, and on page 17, “curve ”should be used in the plural form, suggesting that the author should check carefully the spelling and grammatical errors.

Response: On the basis of your advice, the text was carefully checked for singular and plural errors and word repetition, and other spelling and grammatical errors were corrected.

Comment No.4：In the chapter 3.2 of equivalent calibration of cable , for the finite element simulation of the cable, whether the spiral structure of the conductor is retained or the equivalent simplification, please show the specific.

Response: In the chapter 3.2 of equivalent calibration of cable , for the finite element simulation of the cable, the complex spiral structure of the conductor is retained, ensuring that the composite is actually. At the same time, the direct contact type of the conductors of the spiral structure is defined as non-separated for faster solution iterations, and the power wire core as a whole is connected to each other with frictional constraints. At the same time,“ Figure 8. Finite element model”was redrawn to be more clearly described. The textual description of the changes is also marked in red in the text.

Comment No.5：In the lower label of Figure 17, the initial letters of the subheadings in (e) and (f) should be kept in line with the previous ones and only lowercase.

Response: On the basis of your suggestions, in the lower label of Figure 17, the initial letters of (e) and (f) have been changed to lowercase form and marked in red.

Response to the comments of Reviewer #3

Comment No.1：The abbreviation of "Discrete Element Method Multi-Flexible Body Dynamics" in the last paragraph of page 3 should be changed from "EDM-FMBD" to "DEM-FMBD". The manuscript should ensure that the proper nouns and abbreviations of the scientific and technical paper are correct, and it is recommended to double-check other proper nouns.

Response: The correct abbreviation for "Discrete Element Method Multi-Flexible Body Dynamics" has been changed to "DEM-FMBD" according to your suggestions. Also, the usage of other technical terms that appear in the text has been checked to ensure the accuracy of the terminology used in the scientific research paper. Revisions are highlighted in red.

Comment No.2：2.The English language expression is overall more fluent, but there are some minor grammatical errors. On page 16, in analysis (1), the sentence "The cable and the 35th cable clamp..." is not clear enough. On page 17, in analysis (3), the word "perspective" should be used in its adverbial form.

Response: Some minor grammatical errors in the paper have been carefully checked and revised in line with your suggestions. The inaccurate sentence "The cable and the 35th cable clamp..." has been amended to "The cable and the 35th cable clamp are gradually separated, and then, the contact force disappears after complete separation.”，the word "perspective" is a noun in the text and the revised sentence is “from the perspective of simulation and theoretical analysis.” Revisions are highlighted in red.

Comment No.3：3.In chapter 4.1, the "rodrigues formula" is mentioned in equation 5 for the derivation of the dynamics modeling process of large deformation cables. Please give a brief description of its application.

Response: The "rodrigues formula" was briefly described, explaining the scope of its use and justification. The relevant content was added at the bottom of page 10 and marked in red.

Comment No.4：On page 13, why bushings are used between adjacent cable clamps rather than revolute joints? Please briefly discuss the advantages of this choice as well as the rationality.

Response: Compared to conventional rotational constraints, the bushing connections between cable clamps allow for a small amount of displacement in the lateral and longitudinal directions, in addition to mutual rotation. The way in which the bushings are applied matches the actual state of forces on the pins between the cable clamps, while the allowance of restraint speeds up the convergence of the kinetic equations.The relevant descriptions have been modified and marked in red in the manuscript.

Comment No.5：It is recommended to add some updated references from year 2022 and revise the format with reference to the journal’s requirements.

Response: The literature on prefaces to research in related fields in recent years has been carefully read, and the discussion in the first paragraph of the introduction has been supplemented by the literature on large deformation polymorphic studies [7], [8] and [9], and in the second paragraph of the introduction by the literature on cable equivalence analysis studies [14] and [15], and the manuscript has been added to the end of the Reference with these five related literature in the preamble of the field. The format of the references has been modified in accordance with the requirements of the journal.The changes have been marked in red color.

---

## [Decision Letter · Decision Letter 1]

16 Jan 2023

Research on dynamic characteristics of large deformation shearer cable based on absolute node coordinate formulation method

PONE-D-22-31523R1

Dear Dr. Zhang,

We’re pleased to inform you that your manuscript has been judged scientifically suitable for publication and will be formally accepted for publication once it meets all outstanding technical requirements.

Kind regards,

Zilin Gao, Ph.D

Academic Editor

PLOS ONE

Additional Editor Comments (optional):

Reviewers' comments:

Reviewer's Responses to Questions

**Comments to the Author**

1. If the authors have adequately addressed your comments raised in a previous round of review and you feel that this manuscript is now acceptable for publication, you may indicate that here to bypass the “Comments to the Author” section, enter your conflict of interest statement in the “Confidential to Editor” section, and submit your "Accept" recommendation.

Reviewer #1: All comments have been addressed

Reviewer #3: (No Response)

2. Is the manuscript technically sound, and do the data support the conclusions?

Reviewer #1: Yes

Reviewer #3: (No Response)

3. Has the statistical analysis been performed appropriately and rigorously? 

Reviewer #1: Yes

Reviewer #3: (No Response)

4. Have the authors made all data underlying the findings in their manuscript fully available?

Reviewer #1: Yes

Reviewer #3: (No Response)

5. Is the manuscript presented in an intelligible fashion and written in standard English?

Reviewer #1: Yes

Reviewer #3: (No Response)

6. Review Comments to the Author

Reviewer #1: The manuscript entitled ‘Research on dynamic characteristics of large deformation shearer cable based on absolute node coordinate formulation method’ applies ANCF（Absolute Node Coordinate Formulation） method to study the dynamic characteristics of cable with large deformation.

After revisions, the reviewer considers that the paper has been improved considerably.

In the reviewer's opinion, the paper is ready for publication.

The reviewer would like to congratulate the authors on the effort made to improve the paper.

Reviewer #3: (No Response)

7. PLOS authors have the option to publish the peer review history of their article (what does this mean?). If published, this will include your full peer review and any attached files.

Reviewer #1: No

Reviewer #3: No

---

## [Editor Report · Acceptance letter]

2 Feb 2023

PONE-D-22-31523R1 

*Research on dynamic characteristics of large deformation shearer cable based on absolute node coordinate formulation method*

Dear Dr. Zhang:

I'm pleased to inform you that your manuscript has been deemed suitable for publication in PLOS ONE. Congratulations! Your manuscript is now with our production department. 

Kind regards, 

on behalf of

Dr. Zilin Gao 

Academic Editor

PLOS ONE